# Model Inversion Networks
# for Model-Based Optimization

**Aviral Kumar, Sergey Levine**
Electrical Engineering and Computer Sciences, UC Berkeley
`aviralk@berkeley.edu`

## Abstract

This work addresses data-driven optimization problems, where the goal is to find an input that maximizes an unknown score or reward function given access to a dataset of inputs with corresponding scores. When the inputs are high-dimensional and valid inputs consistute a small subset of this space (e.g., valid protein sequences or valid natural images), such model-based optimization problems become exceptionally difficult, since the optimizer must avoid out-of-distribution and invalid inputs. We propose to address such problem with *model inversion networks* (MINs), which learn an inverse mapping from scores to inputs. MINs can scale to high-dimensional input spaces and leverage offline logged data for both contextual and non-contextual optimization problems. MINs can also handle both purely offline data sources and active data collection. We evaluate MINs on high-dimensional model-based optimization problems over images, protein designs, and neural network controller parameters, and bandit optimization from logged data.

## 1 Introduction

Data-driven optimization problems arise in a range of domains: from protein design [4] to automated aircraft design [13], from the design of robots [18] to the design of neural network architectures [43]. Such problems require optimizing unknown score functions using datasets of input-score pairs, without direct access to the score function being optimized. This can be especially challenging when valid inputs lie on a low-dimensional manifold in the space of all inputs, such as the space of valid aircraft designs or valid images. Existing methods to solve such problems often use derivative-free optimization [34]. Most of these techniques require *active* data collection, where the unknown function is queried at new inputs. However, when function evaluation involves a complex real-world process, such as testing a new aircraft design or evaluating a new protein, such active methods can be very expensive. On the other hand, in many cases there is considerable prior data – existing aircraft and protein designs, and advertisements and user click rates – that could be leveraged to solve the optimization problem.

In this work, our goal is to develop a method to solve such optimization problems that can (1) readily operate on high-dimensional inputs comprising a narrow, low-dimensional manifold in the input space, (2) readily utilize offline static data, and (3) learn with minimal active data collection if needed. We can define this problem setting formally as the optimization problem

$$\mathbf{x}^{\star} = \arg \max_{\mathbf{x}} f(\mathbf{x}), \qquad (1)$$

where the score/reward function $f(\mathbf{x})$ is unknown, and we have access to a dataset $\mathcal{D} = \{(\mathbf{x}_1, y_1), \ldots, (\mathbf{x}_N, y_N)\}$, where $y_i$ denotes the value $f(\mathbf{x}_i)$. In some cases, which we refer to as *active model-based optimization*, we can collect additional data with active queries. However, in many practical settings, no further data collection is possible. We call this *data-driven model-based*

*optimization*. This problem statement can also be extended to the *contextual* setting, where the aim is to optimize the expected score function across a context distribution. That is,

$$\pi^\star = \arg\max_\pi \mathbb{E}_{c\sim p(\cdot)}[f(c, \pi(c))], \tag{2}$$

where $\pi^\star$ maps contexts $c$ to inputs $\mathbf{x}$, such that the expected score under the context distribution $p(c)$ is optimized. As before, $f(c, \mathbf{x})$ is unknown, and we use a dataset $\mathcal{D} = \{(c_i, \mathbf{x}_i, y_i)\}_{i=1}^N$, where $y_i$ is the value of $f(c_i, \mathbf{x}_i)$. Such contextual problems with logged datasets have been studied in the context of contextual bandits [37, 15].

A simple way to approach these model-based optimization problems is to train a proxy function $f_\theta(\mathbf{x})$ or $f_\theta(c, \mathbf{x})$, with parameters $\theta$, to approximate the true score, using the dataset $\mathcal{D}$. However, directly using $f_\theta(\mathbf{x})$ in place of the true function $f(\mathbf{x})$ in Equation (1) generally works poorly, because the optimizer will quickly find an input $\mathbf{x}$ for which $f_\theta(\mathbf{x})$ outputs an erroneously large value. This issue is especially severe when the inputs $\mathbf{x}$ lie on a narrow manifold in a high-dimensional space, such as the set of natural images [42]. The function $f_\theta(\mathbf{x})$ is only valid near the training distribution, and can output erroneously large values when queried at points chosen by the optimizer. Prior work has sought to addresses this issue by using uncertainty estimation and Bayesian models [35] for $f_\theta(\mathbf{x})$, as well as active data collection [34]. However, explicit uncertainty estimation is difficult when the function $f_\theta(\mathbf{x})$ is very complex or when $\mathbf{x}$ is high-dimensional.

Instead of learning $f_\theta(\mathbf{x})$, we propose to learn the inverse function, mapping from values $y$ to corresponding inputs $\mathbf{x}$. This inverse mapping is one-to-many, and therefore requires a *stochastic* mapping, which we can express as $f_\theta^{-1}(y, \mathbf{z}) \to \mathbf{x}$, where $\mathbf{z}$ is a random variable. We term such models *model inversion networks* (MINs). MINs can handle high-dimensional input spaces such as images, can tackle contextual problems, and can accommodate both static datasets and active data collection. We discuss how to design active data collection methods for MINs, leverage advances in deep generative modeling [10, 2], and scale to very high-dimensional input spaces. We experimentally demonstrate MINs in a range of settings, showing that they outperform prior methods on high-dimensional input spaces, such as images, neural network parameters, and protein designs, and substantially outperform prior methods on contextual bandit optimization from logged data [37].

## 2   Related Work

**Bayesian and model-based optimization.** Most prior work on model-based optimization focuses on the active setting. This includes the cross entropy method (CEM) and related derivative-free methods [27, 28], reward weighted regression [25], and Bayesian optimization methods based on Gaussian processes [33, 34, 35]. Many of these methods work best on low-dimensional tasks, where GPs can be trained effectively, while MINs scale to high-dimensional inputs, such as images and protein sequences. More recent methods also replace GPs with parametric acquisition function approximators, such as Bayesian neural networks [35] and latent variable models [16, 8, 7], as well as more recent methods such as CbAS [4]. However, these methods still require querying the true function $f(\mathbf{x})$ at each iteration to iteratively arrive at a near-optimal solution. We show in Section 3.3 that MINs can be applied to such an active setting as well, and in our experiments we show that MINs can perform competitively with these prior methods. However, we mainly focus on the data-driven or static setting, where a dataset is provided in advance, and prior methods are not applicable.

**Contextual bandits.** Equation 2 describes contextual bandit problems. Prior work on batch contextual bandits has focused on batch learning from bandit feedback (BLBF), where the learner needs to produce the best possible policy that optimizes the score function from logged experience. Existing approaches build on the counterfactual risk minimization (CRM) principle [37, 38], and have been extended to work with deep nets [15]. In our comparisons, we find that MINs substantially outperform these prior methods in the batch contextual bandit setting.

**Deep generative modeling.** Recently, deep generative modeling approaches have been very successful at modelling high-dimensional manifolds, such as natural images [10, 39, 6], speech [40], text [41], alloy composition prediction [23]. While such methods can effectively model the data distribution and serve as the basis of our approach, by themselves generative models do not solve the optimization problem, since generating the *best* datapoint necessarily requires stepping outside of the distribution of training data. MINs combine the strength of deep generative models with important algorithmic choices (discussed in Section 3.3, 3.2) to solve model-based optimization problems. We show experimentally that these choices are important for adapting generative models to model-based optimization, and it is difficult to perform effective optimization without them.

# 3 Model Inversion Networks

We now describe our model inversion networks (MINs) method, which can perform both active and data-driven (passive) model-based optimization over high-dimensional inputs.

**Problem statement.** Our goal is to solve optimization problems of the form $\mathbf{x}^\star = \arg\max_\mathbf{x} f(\mathbf{x})$, where the function $f(\mathbf{x})$ is not known, but we must instead use a dataset of input-output tuples $\mathcal{D} = \{(\mathbf{x}_i, y_i)\}$. In the contextual setting described in Equation (2), each datapoint is also associated with a context $c_i$. For clarity, we present our method in the non-contextual setting, but the contextual setting can be derived analogously by conditioning all learned models on the context. In the *active* setting, which is most often studied in prior work, the algorithm can query $f(\mathbf{x})$ few times on each iteration to augment the dataset, while in the *static* or *data-driven* setting, only an initial static dataset is available. The goal is to obtain the best possible $\mathbf{x}^\star$, with the highest possible value of $f(\mathbf{x}^\star)$.

One naïve way of solving MBO problems is to learn a proxy score function $f_\theta(\mathbf{x})$ via empirical risk minimization, and then maximize it with respect to $\mathbf{x}$. However, naïve applications of such a method would fail for two reasons. First, the proxy function $f_\theta(\mathbf{x})$ may not be accurate outside the distribution on which it is trained, and optimization with respect to it may simply lead to values of $\mathbf{x}$ for which $f_\theta(\mathbf{x})$ makes the largest mistake. The second problem is more subtle. When $\mathbf{x}$ lies on a narrow manifold in a very high-dimensional space, such as the space of natural images, the optimizer can produce invalid values of $\mathbf{x}$, which result in arbitrary outputs when fed into $f_\theta(\mathbf{x})$. Since the shape of this manifold is unknown, it is difficult to constrain the optimizer to prevent this. This second problem is rarely addressed or discussed in prior work, which typically focuses on optimization over low-dimensional and compact domains with known bounds. We will show in our experiments in Section 4, that optimizing $\mathbf{x}$ with respect to a proxy function $f_\theta(\mathbf{x})$ fails in several high-dimensional settings, often going off the manifold (Figure 2).

## 3.1 Optimization via Inverse Maps

Part of the reason for the brittleness of the naïve approach above is that $f_\theta(\mathbf{x})$ has a high-dimensional input space, making it easy for the optimizer to find inputs $\mathbf{x}$ for which the proxy function produces an unreasonable output. Can we instead learn a function with a small input space, which implicitly understands the space of valid, in-distribution values for $\mathbf{x}$? The main idea behind our approach is to learn an inverse map that produces a value of $\mathbf{x}$ given a score value $y$, given by $f_\theta^{-1} : \mathcal{Y} \to \mathcal{X}$. The input to the inverse map is a scalar, making it comparatively easy to constrain to valid values, and by directly generating the inputs $\mathbf{x}$, an approximation to the inverse function must implicitly understand which input values are valid. As multiple $\mathbf{x}$ values can correspond to the same $y$, we design $f_\theta^{-1}$ as a stochastic map that maps a score value along with a $d_z$-dimensional random vector to a $\mathbf{x}$, $f_\theta^{-1} : \mathcal{Y} \times \mathcal{Z} \to \mathcal{X}$, where $\mathbf{z}$ is distributed according to a prior distribution $p_0(\mathbf{z})$. The inverse map training objective corresponds to standard distribution matching, analogously to generative models, which we will express in a somewhat more general way to simplify the exposition later. Let $p_\mathcal{D}(\mathbf{x}, y)$ denote the data distribution, such that $p_\mathcal{D}(y)$ is the marginal over $y$, and let $p(y)$ be an any distribution on $\mathcal{Y}$, which could be equal to $p_\mathcal{D}(y)$. We can train the proxy inverse map $f_\theta^{-1}$ by minimizing the following objective:

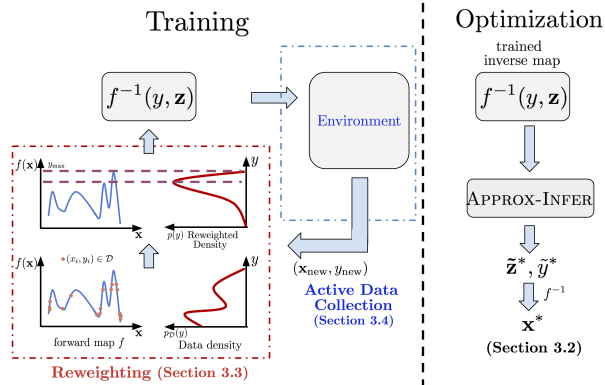

Figure 1: Schematic for MIN training and optimization. Reweighting (Section 3.3) and, optionally, active data collection (Section 3.4) are used during training. The MIN is then used to obtain the optimal input $\mathbf{x}^\star$ using the Approx-Infer procedure in Section 3.2.

$$\mathcal{L}_p(\mathcal{D}) = \mathbb{E}_{y \sim p(y)} \left[ D\left( p_\mathcal{D}(\mathbf{x}|y), p_{f_\theta^{-1}}(\mathbf{x}|y) \right) \right], \tag{3}$$

where $p_{f_\theta^{-1}}(\mathbf{x}|y)$ is obtained by marginalizing over $\mathbf{z}$, and $D$ is a measure of divergence between the two distributions. Using the Kullback-Leibler divergence leads to maximum likelihood learning, while Jensen-Shannon divergence leads to a GAN-style training objective. MINs can be adapted

to the contextual setting by passing in the context as an input and learning $f_\theta^{-1}(y_i, z, c_i)$. While the basic idea behind MINs is simple, a number of implementation choices are important for good performance. Instead of choosing $p(y)$ to be $p_\mathcal{D}(y)$, as in standard ERM, in Section 3.3 we show that a careful choice of $p(y)$ leads to better performance. In Section 3.4, we then describe a method to perform active data sampling with MINs. We also present a method to generate the best optimization output $\mathbf{x}^\star$ from a trained inverse map $f_\theta^{-1}(\cdot, \cdot)$ during evaluation in Section 3.2. The structure of the full MIN algorithm is shown in Algorithm 1, and a flowchart of the procedure is shown in Figure 1.

## 3.2 Inference with Inverse Maps (Approx-Infer)

Once the inverse map is trained, the goal of our algorithm is to generate the best possible $\mathbf{x}^\star$, which will maximize the true score function as well as possible. Since a score $y$ needs to be provided as input to the inverse map, we must carefully the score $y$ with which to query the inverse map to obtain the best $\mathbf{x}$. One naïve

---
**Algorithm 1** Generic Algorithm for MINs

1: Train inverse map $f_\theta^{-1} : \mathcal{Y} \times \mathcal{Z} \to \mathcal{X}$ with Equation (3)
2: (optionally) perform active data collection
3: return $\mathbf{x}^\star \leftarrow \textsc{Approx-Infer}\ (f_\theta^{-1}, p_\mathcal{D}(y))$

---

heuristic is to pick the best $y_{\max} \in \mathcal{D}$ and produce $\mathbf{x}_{\max} \sim f_\theta^{-1}(y_{\max}^*)$ as the output. However, a data-driven MBO method should be able to extrapolate beyond the best score in the dataset.

In order to extrapolate as far as possible, while still staying on the data manifold, we need to measure the validity of the generated samples $\mathbf{x}$. One way to do this is to measure the agreement between the learned inverse map and an independently trained forward model $f_\theta$: the values of $y$ for which the generated samples $\mathbf{x}$ are predicted to have a score similar to $y$ are likely in-distribution, whereas those where the forward model predicts a very different score may be too far outside the training distribution. This amounts to using the agreement between independently trained forward and inverse maps to quantify the degree to which a particular score $y$ is out-of-distribution. Since the latent variable $z$ captures the multiple possible outputs of the one-to-many inverse map, we can further optimize over $\mathbf{z}$ for a given $y$ to find the best, most trustworthy $\mathbf{x}$ subject to the constraint that $\mathbf{z}$ has a high likelihood under the prior. This can be formalized as the following optimization:

$$\tilde{y}^*, \tilde{\mathbf{z}}^* := \arg\max_{y, \mathbf{z}} f_\theta(f_\theta^{-1}(\mathbf{z}, y)) \ \text{ s.t. } \ ||y - f_\theta(f_\theta^{-1}(\mathbf{z}, y))||_2^2 \le \epsilon_1, \ \ p(\mathbf{z}) \ge \epsilon_2$$

In order to perform this optimization, we can perform gradient descent on $y$ and $\mathbf{z}$ so as to minimize the corresponding Lagrangian. More details on solving the Lagrangian are provided in Appendix A. This optimization can be motivated as finding an extrapolated score, higher than the best score in the dataset $\mathcal{D}$, that corresponds to values of $\mathbf{x}$ that lie on the valid input manifold, and for which independently trained forward and inverse maps agree. Although this optimization uses an approximate forward map $f_\theta(\mathbf{x})$, we show in our experiments in Section 4 that it produces substantially better results than optimizing with respect to a forward model alone. The inverse map substantially constraints the search space, requiring an optimization over a 1-dimensional $y$ and a (relatively) low-dimensional $z$, rather than the full space of inputs. This can be viewed as a special (deterministic) case of a probabilistic optimization procedure, which we describe and analyze in Appendix A.

## 3.3 Reweighting the Training Distribution

A naïve implementation of the training objective in Equation (3) samples $y$ from the data distribution $p_\mathcal{D}(y)$ during training. However, since our goal is to perform optimization, it is much more important for the inverse map to produce accurate generations for *high* values of $y$ rather than low values. We could therefore consider a weighted training objective, with higher weights on points with larger scores. We could train only on the best points – either the single datapoint with the largest $y$ or, in the contextual case, the points with the largest $y$ for each context. However, training on only a single datapoint gives rise to a very high-variance training objective. We can instead train on a distribution of $y$ values, $p(y)$, that is supported on the largest number of near-optimal $y$ values. This amounts to trading off the bias induced by not training on the optimal $y^\star$ alone, and variance induced by training on only a few samples with high $y$ values. Theorem 3.1 illustrates this bias-variance tradeoff.

In order to train under a particular distribution $p(y)$, we re-weight the training points using importance sampling. A datapoint $(\mathbf{x}_i, y_i)$ sampled from $p_\mathcal{D}$ is assigned an importance weight $\mathbf{w}_i = \frac{p(y_i)}{p_\mathcal{D}(y_i)}$. The reweighted objective is given by $\hat{\mathcal{L}}_p(\mathcal{D}) = \frac{1}{|\mathcal{D}|} \sum_i \mathbf{w}_i \cdot \hat{D}(\mathbf{x}_i, f_\theta^{-1}(y_i))$. By minimizing the bound

in Theorem 3.1, we obtain $p(y) \propto \frac{N_y}{N_y + K} \cdot \exp(y - y^\star)$, where $N_y$ is equal to the number of points $(\mathbf{x}, y)$ in $\mathcal{D}$ with score equal to $y$, and $y^\star$ is the highest score in $\mathcal{D}$. This is consistent with intuition: our objective upweights points with high $y$-values, provided the number of samples at that score, $N_y$, is not too low. For continuous-valued scores, we bin the $y$-values into discrete bins for reweighting, as we discuss in Section 3.5. We now present the bias-variance tradeoff result that motivates this choice by bounding the variance and bias of $\nabla_\theta \hat{\mathcal{L}}_p(\mathcal{D})$:

**Theorem 3.1** (Bias + variance bound in MINs). *Let $\mathcal{L}(p^*)$ be the expected objective value under the Dirac-delta distribution, $p^*(y) = \mathbf{1}[y = y^*]$, centered at the optimal $y^* \in \mathcal{D}$: $\mathcal{L}(p^*) = \mathbb{E}_{y \sim p^*(y)}[D(p(\mathbf{x}|y), f^{-1}(y))]$. Let $N_y$ be the number of datapoints with the particular $y$ value observed in $\mathcal{D}$, and $d_2$ denote the exponentiated Renyi divergence For some constants $C_1, C_2, C_3$, with high confidence,*

$$\mathbb{E}\left[ ||\nabla_\theta \hat{\mathcal{L}}_p(\mathcal{D}) - \nabla_\theta \mathcal{L}(p^*)||_2^2 \right] \leq C_1 \mathbb{E}_{y \sim p(y)}\left[ \frac{1}{N_y} \right] + C_2 \frac{d_2(p||p_\mathcal{D})}{|\mathcal{D}|} + C_3 \cdot D_{\mathrm{TV}}(p^*, p)^2.$$

Theorem 3.1 suggests a tradeoff between being close to the optimal $y^*$ (third term) and reducing variance by covering the full data distribution $p_\mathcal{D}$ (second term), and our choice of $p(y)$ defined previously minimizes this bound. A proof of Theorem 3.1, along with an derivation for why our choice of $p(y)$ minimizes the bias-variance tradeoff is provided in Appendix B.

### 3.4 Active Data Collection via Randomized Labeling

Effectively using MINs in the active setting also requires an intelligent exploration strategy for selecting the query point $\mathbf{x}_t^\star$ at each iteration. While we could select this point greedily, by using the procedure in Section 3.2, previous work on active MBO (e.g., Bayesian optimization) has observed that uncertainty-aware exploration can be substantially more effective [4, 30, 31]. To that end, we propose a novel approach for exploration with MINs that resembles Thompson sampling [30]. A full derivation of this method, which includes a proof bounding the regret under some mild assumptions, is presented in Appendix C due to space constraints. Here we briefly describe the approach in Algorithm 2. Our aim is to approximately sample from the posterior over inverse maps $f_{\theta_t}^{-1}$ at each iteration, given the dataset $\mathcal{D}_t$ available at that iteration, and then select $\mathbf{x}_t^\star$ as the optimum of this sampled function. As we show in Appendix C, we can approximate such posterior samples by training on an *augmented* dataset $\mathcal{D}'_t = \mathcal{D}_t \cup \mathcal{S}_t$, where $\mathcal{S}_t = \{(\tilde{\mathbf{x}}_j, \tilde{y}_j)\}_{j=1}^K$ is a dataset of synthetically generated input-score pairs corresponding to unseen $y$ values in $\mathcal{D}_t$. While there are many ways to generate these points, we simply sample $\mathbf{x}$ values randomly from $\mathcal{D}_t$, and sample $y$ near-optimal observed $y$ values biased with additive noise.

---

**Algorithm 2** Active MINs with Randomized Labeling

1: Initialize inverse map, $f_\theta^{-1}$, dataset $\mathcal{D}_0 = \{\}$,
2: **for** step $t$ in $\{0, \ldots, T\text{-}1\}$ **do**
3:     Sample synthetic samples $\mathcal{S}_t = \{(\mathbf{x}_i, y_i)\}_{i=1}^K$)
4:     Train *inverse map* $f_t^{-1}$ on $\mathcal{D}'_t = \mathcal{D}_t \cup \mathcal{S}_t$, using reweighting described in Section 3.3
5:     Query function $f$ at $\mathbf{x}_t^\star = f_t^{-1}(\max_{\mathcal{D}'_t} y)$
6:     Observe $(\mathbf{x}_t^\star, f(\mathbf{x}_t^\star))$ and update $\mathcal{D}_{t+1} = \mathcal{D}_t \cup (\mathbf{x}_t^\star, f(\mathbf{x}_t^\star))$
7: **end for**

---

### 3.5 Practical Implementation of MINs and Algorithm Summary

We now describe our instantiation of MINs for high-dimensional inputs with deep neural network models. The inverse map in MINs must model the manifold of valid, high-dimensional $\mathbf{x}$, making GANs [10] a suitable choice. We instantiate our inverse map with a GAN by choosing $D$ in Equation 3 to be the *Jensen-Shannon* divergence measure. Since we generate $\mathbf{x}$ conditioned on $y$, the discriminator is parameterized as $\mathrm{Disc}(\mathbf{x}|y)$, and trained to output 1 for a valid $(\mathbf{x}, y)$ pair (i.e., where $y = f(\mathbf{x})$ and $\mathbf{x}$ comes from the data) and 0 otherwise. Thus, we optimize the following objective:

$$\min_\theta \max_{\mathrm{Disc}} \mathcal{L}_p(\mathcal{D}) = \mathbb{E}_{y \sim p(y)}\big[\mathbb{E}_{\mathbf{x} \sim p_\mathcal{D}(\mathbf{x}|y)}[\log \mathrm{Disc}(x|y)] + \mathbb{E}_{\mathbf{z} \sim p_0(\mathbf{z})}[\log(1 - \mathrm{Disc}(f_\theta^{-1}(\mathbf{z}, y)|y))]\big].$$

This model is similar to a conditional GAN (cGAN), which has been used in the context of modeling distribution of $\mathbf{x}$ conditioned on a discrete-valued label [22]. As discussed in Section 3.3, we additionally reweight the data distribution using importance sampling. To that end, we discretize the space $\mathcal{Y}$ into $B$ discrete bins $b_1, \cdots, b_B$ and, following Section 3.3, weight each bin $b_i$ according to $p(b_i) \propto \frac{N_{b_i}}{N_{b_i} + \lambda} \exp\left( \frac{-|b_i - y^*|}{\tau} \right)$, where $N_{b_i}$ is the number of datapoints in the bin, $y^*$ is the maximum score observed, and $\tau$ is a hyperparameter. More details are given in Appendix D. In the active setting,

| Dataset & Type | BanditNet | BanditNet$^*$ | MIN w/o I | MIN (Ours) | MINs w/o R |
|---|---|---|---|---|---|
| MNIST (49% corr.) | $36.42 \pm 0.6$ | $-$ | $94.2 \pm 0.13$ | $\mathbf{95.0 \pm 0.16}$ | $95.0 \pm 0.21$ |
| MNIST (Uniform) | $9.94 \pm 0.0$ | $-$ | $92.21 \pm 0.22$ | $\mathbf{93.67 \pm 0.51}$ | $92.8 \pm 0.01$ |
| CIFAR-10 (49% corr.) | $42.13 \pm 2.35$ | $87.0$ | $91.35 \pm 0.87$ | $\mathbf{92.21 \pm 1.0}$ | $89.02 \pm 0.05$ |
| CIFAR-10 (Uniform) | $14.43 \pm 1.43$ | $-$ | $76.31 \pm 0.40$ | $\mathbf{77.12 \pm 0.54}$ | $74.87 \pm 0.12$ |

Table 1: Test accuracy on MNIST and CIFAR-10 with 50k bandit feedback training examples. BanditNet$^*$ is the result from Joachims et al. [15], while the BanditNet column is our implementation; we were unable to fully replicate the performance from prior work (details in Appendix D). MINs outperform BanditNet and BanditNet$^*$, both with and without the inference procedure in Section 3.2. MINs w/o reweighting perform on par with full MINs on MNIST, and slightly worse on CIFAR 10, while still outperforming the prior method.

we perform active data collection using the randomized labelling algorithm described in Section 3.4. We provide the implementation details for randomized labeling in Appendix D.

## 4 Experimental Evaluation

The goal of our empirical evaluation is to answer the following questions: **(1)** Can MINs successfully solve optimization problems of the form shown in Equations 1 and 2, in both static and active settings, more effectively than prior methods? **(2)** Can MINs generalize to high dimensional spaces, where valid inputs $\mathbf{x}$ lie on a lower-dimensional manifold, such as the space of images? **(3)** How do the different components of our method, including reweighting, our inference procedure, and (in the active setting) our exploration method, influence performance? To this end, we evaluate MINs on a wide range of tasks, including batch contextual bandits for image classification and optimization over hand-written character images, faces, protein designs and neural network parameters. Note that very few prior works consider the static or data-driven MBO setting, limiting the set of possible comparisons. We compare to the BanditNet model [15] in the contextual case, and a standard forward model on images, while including more prior methods in the active setting.

### 4.1 Data-Driven Optimization with Static Datasets

We first study the *data-driven* optimization setting, where the goal is to find an input $\mathbf{x}$ that achieves a better score than any point in the training set. We evaluate our method on a batch contextual bandit task proposed in prior work [15], optimization over natural images, and optimization over the space of neural network parameters for controller optimization.

**Batch contextual bandits.** We first study the contextual optimization problem described in Equation 2. The goal is to learn a policy, purely from static data, to maximize the score $f(c, \pi(c))$ on average across contexts from $p(c)$. Following Joachims et al. [15], we evaluate learned policies trained on a static dataset for simulated classification tasks on MNIST and CIFAR-10. These tasks formulate a standard classification problem on MNIST or CIFAR-10 as a contextual bandit optimization problem, where the context space is the set of all images in the MNIST or CIFAR dataset and the action space is given by the discrete space over the ten possible labels. However, unlike standard supervised learning, instead of providing the actual label for each image, we construct an optimization dataset by selecting images from the MNIST or CIFAR dataset as the context $c$, a random label as the *input* $\mathbf{x}$, and a binary indicator indicating whether or not the label is correct as the *score* $y$. The goal is to maximize the expected value of the binary indicator (i.e., the score $y$) on the test dataset of unseen contexts, which is equivalent to the accuracy of the learned model on the test set. Note that we do not use any sort of surrogate objective (e.g., log-likelihood), since we are not provided with the correct label for each image.

We evaluate on two datasets, which are formed by: (1) selecting random labels $\mathbf{x}_i$ for each context $c_i$; (2) selecting the correct label 49% of the time, which matches the protocol in [15]. In Table 1, we report the average score on unseen "test" contexts, which corresponds to the test set accuracy of the model as discussed above. We compare our method to the previously proposed BanditNet model [15]. The results show that MINs outperform BanditNet on both MNIST and CIFAR-10, indicating that MINs can successfully perform contextual model-based optimization in the data-driven setting.

**Character stroke width optimization.** In the next experiment, we study how well MINs optimize over high-dimensional inputs, where valid inputs lie on a lower-dimensional manifold. We first construct an image optimization task out of the MNIST [17] dataset. The goal is to optimize *directly* over $\mathbf{32{\times}32 = 1024}$ image pixels, to produce images with the thickest stroke width, such that the image corresponds, in the first scenario, (a) to any valid character, and in the second scenario, (b), to a valid instance of a particular character class (3 in this case). That is, the goal is to produce the

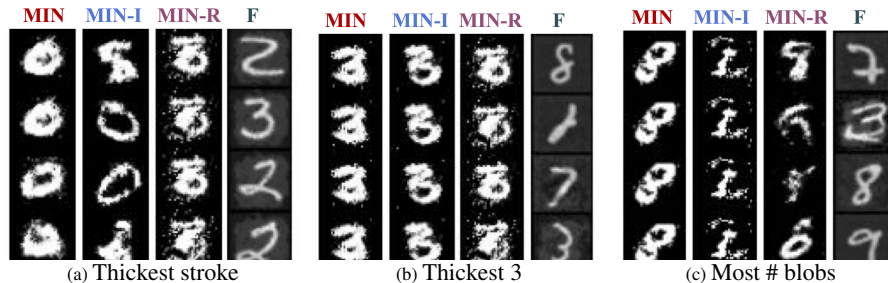

| (a) Thickest stroke | (b) Thickest 3 | (c) Most # blobs |

Figure 2: Results for non-contextual data-driven optimization on MNIST: (a), (b): Stroke width optimization, and (c): Maximization of disconnected black pixel blobs. From left to right: MINs, MINs w/o Inference (Sect 3.2), MINs w/o reweighting (Sec 3.3), and direct optimization of a forward model (F). MINs produce the thickest characters that still resemble valid digits. Optimizing the forward function directly simply brightens the background. Quantitative scores in Table 2. The optimal answer in (b) is to produce the thickest 3, and in (c) is to produce the thickest 8.

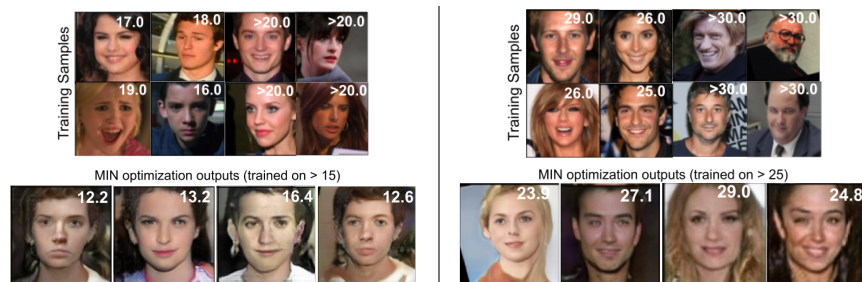

Figure 3: MIN optimization to obtain the youngest faces ($\mathbf{x}$) when trained on faces older than 15 (left) and older than 25 (right). Samples from each training set are shown in the top row. Optimized images $\mathbf{x}$ (bottom) are obtained via inference at different points during model training. We overlay the actual negative score (age) for each face on the real images, and the age obtained from subjective user rankings on the results.

thickest character that is still recognizable. In both scenarios (a) and (b), we define stroke width as the number of pixels with intensity more than a threshold (= 0.2) in the image. An image is invalid if the number of pixels with intensity more than 0.2 is more than 90%, indicating that this image is unlike MNIST digits. We also include a third scenario (c), where the goal is to maximize the number of disconnected blobs of black pixels while still keeping the digit recognizable. In Figure 2, we observe that MINs find images $\mathbf{x}$ that maximize the respective score functions. For comparison, we evaluate a method that directly optimizes the image pixels with respect to a forward model, of the form $f_\theta(\mathbf{x})$. This method quickly deviates from the data distribution, simply brightening all pixels in the image.

**Semantic image optimization.** Next, we aim to evaluate MINs on more complex images and score functions, optimizing over color images in a **64×64×3 = 12288**-dimensional space. We consider MBO tasks on the IMDB-Wiki faces [26] dataset, where the function $f(\mathbf{x})$ is the negative of the age of the person in the image. Hence, images with younger faces have higher scores. We construct two versions of this task: (1) training on all faces older than 15, and (2) training of all faces older than 25. This ensures that our model cannot simply copy the youngest face. To obtain ground truth scores for

| Task | Dimensions | Dataset (avg) | Dataset (best) | Forward map | MIN (Ours) |
|------|-----------|---------------|----------------|-------------|------------|
| MINST (a) | 1024 | 149.0 | 265.0 | Invalid | **276.3** |
| MNIST (b) | 1024 | 149.0 | 163.0 | Invalid | **234.3** |
| MNIST (c) | 1024 | 1.6 | **3.0** | 1.4 | 2.3 |
| Faces ($\geq$ 15) | 12288 | 38.7 | -15.0 | Invalid | **-12.2** |
| Faces ($\geq$ 25) | 12288 | 41.5 | -25.0 | Invalid | **-23.9** |
| Hopper | 3843 | 442.9 | 1915.5 | 93.1 | **1960.1** |
| Pendulum | 1537 | 14.7 | 344.5 | 3.4 | **1000.0** |

Table 2: Results for all datasets in the non-contextual data-driven MBO setting. We report the scores for our method (MINs), as well as a forward model ("forward map") baseline, which trains a model of the form $f_\theta(\mathbf{x}) \rightarrow y$ (higher is better). We also indicate the average and best score in the dataset, and the dimensionality of the inputs $\mathbf{x}$. In all cases, MINs outperform the forward model, and produce a point with higher score than the best in the dataset in almost all cases. An "Invalid" score indicates that the model produces inputs $\mathbf{x}$ which not valid digits or faces, going off the manifold of valid inputs.

the generated faces, we use subjective judgement from 13 human subjects, each of whom was asked to answer 35 forced-choice questions indicating which of two faces is younger. We then fit an age to these binary preferences, analogously to Christiano et al. [5].

Following the evaluation protocol in practical MBO problems [1, 4], we report the average and best scores over a set of top-4 optimized images produced by MINs in Table 2. The average age of optimized images, when training MINs with images 15 years or older, was **13.6 years**, with the best image having an age of **12.2**. The model trained on ages 25 and above produced more mixed results, with an average age of **26.2**, and a minimum age of **23.9**. We show the images themselves in Figure 3. A forward model baseline, which directly trains a model of the form $f_\theta(\mathbf{x}) \to y$ and then optimizes w.r.t. $\mathbf{x}$, was unable to produce plausible images. We also compare to a cGAN baseline, which fails to optimize the age, and instead produces faces that resemble random ages from the training set. These results are discussed in Appendix D.1. Note that this task is exceptionally difficult, since the model must extrapolate outside of the ages seen in the dataset, picking up on patterns in the images to produce faces that appear *younger* than any face observed, while avoiding unrealistic images.

**Neural network parameter optimization.** Next, we evaluate MINs for directly optimizing over the parameters of a two-layer, fully-connected ReLU neural network controller on two environments from Gym [3]: Hopper-v2 (**3843** dimensions) and InvertedPendulum-v2 (**1537** dimensions). Note that this is *not* framed as a contextual or reinforcement learning task. MINs directly output the weights of the neural network controller. The goal is to maximize the total reward obtained when the controller is executed in the simulator. We collected a dataset of 552 (Hopper) and 3480 (Pendulum) controller weights and their average returns by using a partial run of the SAC algorithm [12]. The scores in the dataset are computed via noisy evaluation, where a single design can attain different return values due to the stochasticity of the underlying task. Noise in the evaluation function is reflective of stochasticity in the complex real-world evaluation procedure that we would expect a general model-based optimization problem to exhibit. Our model is trained on just this dataset, with no active interaction. Observe in Table 2 that policy parameters produced by MINs obtain a higher score than the best sample in the dataset in both domains. MINs achieve **10x** and **300x** times the performance of the forward model on these tasks.

These results suggest that MINs can outperform previously proposed methods in the data-driven contextual setting, and produce results that exceed the best datapoint in the dataset, as well as the result of the forward model, in the non-contextual setting on very high dimensional tasks.

## 4.2 Ablation Studies

In this section, we present ablation experiments aimed at identifying the benefits of different components of model inversion networks. First, we evaluate MINs when either Approx-Infer or re-weighting are removed. Second, we compare MINs to two ensemble variants of the forward model that incorporate mechanisms to prevent out-of-distribution inputs. These variants serve the same purpose as Approx-Infer in MINs, and hence allow us to isolate and assess the benefits of inverse map modeling.

**Contextual bandits.** The results in Table 1 for bandit optimization show that using Approx-Infer improves test-accuracy by **1.5%** and **1.0%** on MNIST and CIFAR-10, respectively. Utilizing reweighting improves by about **2.5%** on CIFAR-10, indicating that both re-weighting and Approx-Infer are essential.

**Optimization over images (MNIST/ Faces).** With digit images in Figure 2, MINs without Approx-Infer yields invalid solutions **(a, c)** – that do not resemble any digit, and MINs without re-weighting fails to output images belonging to the highest score class **(a, c)** (MINs output a digit "3" as compared to "0", which has the highest thickness in the dataset). Quantitatively, MINs w/o re-weighting obtain **1.18** points lesser and MIN w/o Approx-Infer obtains **1.52** points lesser than MINs in Figure 2(c). Standard cGAN models, which do not utilize re-weighting, shown in Figure 4 in Appendix D.1, output images that resemble random samples from the data distribution.

**Neural network controller.** When learning neural network controllers, MINs without reweighting achieve a return of 951.6 on Hopper (vs. **1960.1** for MINs) and 735.4 on Pendulum (vs. **1000.0** for MINs). We also compare to MINs without Approx-Infer, which achieves a return of 1569.3 and 1000.0 on Hopper and Pendulum respectively which is also lower than MINs.

**Forward models with additional out-of-distribution correction.** Finally, in order to compare the isolated benefits of modeling the inverse map as compared to a forward map, we experimented with

two other variants of forward models on the controller task. These variants aim to prevent the forward model from producing out-of-distribution inputs, serving a similar role to the Approx-Infer procedure in MINs: **(1)** an ensemble of forward models, where we optimize over the mean of the ensemble predictions; **(2)** an ensemble of forward models where the mean predicted value is accompanied with a penalty on the standard deviation of the predictions across the different models in the ensemble during optimization. **(2)** is equivalent to estimating a high-confidence lower-bound estimate of the objective function and optimizing $\mathbf{x}$ against this estimate. We find that **(1)** attains a mean return of **325.3** on the Hopper domain and **(2)** attains **165.3**, which are both higher than the standard forward model (see Table 2), however, still substantially lower than MIN that attains **1960.1** mean return.

All of this evidence indicates the importance of re-weighting and Approx-Infer in MIN optimization.

### 4.3 Optimization with Active Data Collection

In the active MBO setting, MINs must select which *new* datapoints to query to improve their estimate of the optimal input. In this setting, we evaluate MINs on a high-dimensional optimization task over protein sequences, previously used in Brookes et al. [4]. We validate the effectiveness of randomized labeling via a one-dimensional simple didactic visualization (Appendix D.3).

Our high-dimensional active MBO task requires optimizing protein fluorescence over protein designs by selecting variable length sequences of ($\sim$**200**) codons, where each codon can take on one of 20 values, previously used in [4]. This tasks effectively requires us to optimize over around **200 $\times$ 20 = 4000** dimensions. In order to model discrete values, we use a Gumbel-softmax GAN, also previously employed by Gupta and Zou [11]. More details are in Appendix D. Each algorithm is provided with an initial dataset, and then allowed an identical, limited number of score function queries, for which it receives a noisy score value. We use the trained scoring oracles released by [4]. We compare to CbAS [4] and other methods, including CEM (cross entropy method), RWR (reward weighted regression) and forward function – GB [9]. Following [4], we report the ground truth score of a set of 100 optimized outputs (max), and the 50th-percentile score. Note in Table 3 that MINs are comparable to the best performing method on this task, and produce outputs with the highest scores.

| Method | Max | 50%ile |
|---|---|---|
| *MIN (Ours)* | **3.42** | 3.24 |
| *MIN - R* | 3.37 | **3.28** |
| *CbAS* | 3.36 | **3.28** |
| *RWR* | 3.00 | 2.97 |
| *CEM-PI* | 2.92 | 2.9 |
| $GB^*$ | 3.25 | 3.25 |

Table 3: Protein design results, with maximum and the 50th percentile fluorescence out of 100 samples. Prior methods from Brookes et al. [4]. MINs perform comparably to CbAS, producing highest scoring samples.

## 5 Discussion

In this work, we presented a novel approach to model-based optimization (MBO). Instead of learning a proxy forward function $f_\theta(\mathbf{x})$ from inputs $\mathbf{x}$ to scores $y$, MINs learn a stochastic inverse mapping from scores $y$ to inputs. MINs can optimize over high dimensional $\mathbf{x}$ values where valid inputs lie on a narrow manifold, such as in the case of natural images. MINs can optimize effectively in data-driven setting, without active data collection. Our experiments show that MINs are capable of solving MBO tasks in both contextual and non-contextual settings, and can optimize complex score functions, such as the age of a person in a photograph or the performance of a controller defined by neural network weights. Prior work has usually considered MBO in the active setting, where the algorithm can perform a number of active queries. While MINs can handle this setting also, we focus especially on the "data-driven" MBO problem. This is particularly important when data collection is expensive or unsafe, but prior data is available for optimization, as in the case of medical applications such as drug design, or engineering applications such as aircraft design. We believe that this problem setting is very important, and worth exploring further in future work. A number of important open problems remain. For example, can we characterize the space of data-driven MBO problems where extrapolation beyond the training set is viable? Can we provide calibrated confidence bounds? Can we better exploit problem structure when it is available? Answers to these questions in future work may further expand the range of data-driven MBO problems that can be addressed.

### Broader Impact

In this work we introduced model-inversion networks (MIN), a novel approach to model-based optimization, which can be utilized for solving both passive "data-driven" optimization problems

and active model-based optimization problems. Model-based optimization is a generic black-box optimization framework that captures a wide range of practically interesting and highly relevant optimization problems, such as drug discovery, controller design, and optimization in computer systems. In this work, we demonstrate the efficacy of MINs in high dimensional optimization problems, from complex input types (raw pixels, raw neural network weights, protein sequences).

We are particularly excited about application of MINs in the domain of drug design and discovery, and other computational biology domains. The importance of the problem of drug design needs no motivation or justification, especially in these times of this pandemic that mankind is facing now. Existing methods in place for these problems typically follow an "active" experimental pipeline – the designs proposed by an ML or computational model are evaluated in real life, and then the results of these evaluations are incorporated into further model training or model improvement. Often the evaluation phase is the bottleneck: this phase is highly expensive both in terms of computational resources and time, often requiring human intervention, and in some cases, taking months of time. We can avoid these bottlenecks by instead solving such optimization problems to the best possible extent in the static data-driven setting, by effectively reusing both good and bad data from past experiments, and MINs are designed to be efficient at exactly this. Beyond the field of biology, there are several other applications for which our proposed method is relevant. Design problems in engineering, such as design of aircraft, are potential applications of our method. There are also likely to be applications in computer systems and architectures.

While effective model-based optimization algorithms can have considerable positive economic and technological growth effects, it can also enable applications with complex implications with regards to safety and privacy (for example, safety issues in drug design or aircraft design, or privacy issues in computer systems optimization), as well as in terms of complex economic effects for example, changing job requirements and descriptions, due to automation of certain tasks. Both of these implications are not unique to our method, but more generally apply to situations where black box autonomous neural network models are deployed in practice.

## Acknowledgements and Funding

We thank all memebers of the Robotic AI and Learning Lab at UC Berkeley for their participation in the human study. We thank anonymous reviewers for feedback on an earlier version of this paper. This research was funded by the DARPA Assured Autonomy program, the National Science Foundation under IIS-1700697, and compute support from Google, Amazon, and NVIDIA.

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
