[Supplementary Material]

# Appendices

## A   Probabilistic Interpretation of Section 3.2

In this section, we show that the inference scheme described in Equation 4, Section 3.2 emerges as a deterministic relaxation of the probabilistic inference scheme described below. We reiterate that in Section 3.2, a singleton $\mathbf{x}^*$ is the output of optimization, however the procedure can be motivated from the perspective of the following probabilistic inference scheme.

Let $p(\mathbf{x}|y)$ denote a stochastic inverse map, and let $p_f(y|\mathbf{x})$ be a probabilistic forward map. Consider the following optimization problem:

$$\arg\max_{y,\hat{p}} \ \mathbb{E}_{\mathbf{x}\sim\hat{p}(\mathbf{x}|y),\hat{y}\sim p_f(\hat{y}|\mathbf{x})}[\hat{y}] \ \text{ such that } \ \mathcal{H}(\hat{y}|\mathbf{x}) \leq \epsilon_1, D(\hat{p}(\mathbf{x}|y), p_\theta(\mathbf{x}|y)) \leq \epsilon_2, \qquad (4)$$

where $p_\theta(\mathbf{x}|y)$ is the probability distribution induced by the learned inverse map (in our case, this corresponds to the distribution of $f_\theta^{-1}(y, z)$ induced due to randomness in $z \sim p_0(\cdot)$), $p_f(\mathbf{x}|y)$ is the learned forward map, $\mathcal{H}$ is Shannon entropy, and $D$ is KL-divergence measure between two distributions. In Equation 4, maximization is carried out over the input $y$ to the inverse-map, and the input $z$ which is captured in $\hat{p}$ in the above optimization problem, i.e. maximization over $z$ in Equation 4 is equivalent to choosing $\hat{p}$ subject to the choice of singleton/ Dirac-delta $\hat{p}$. The Lagrangian is given by:

$$\mathcal{L}(y, \hat{p}; p, p_f) = \mathbb{E}_{\mathbf{x}\sim\hat{p}(\mathbf{x}|y),\hat{y}\sim p_f(\hat{y}|\mathbf{x})}[\hat{y}] + \lambda_1 \left(\mathbb{E}_{\mathbf{x}\sim\hat{p}(\mathbf{x}|y),\hat{y}\sim p_f(\hat{y}|\mathbf{x})}[\log p_f(\hat{y}|\mathbf{x})] + \epsilon_1\right) +$$
$$\lambda_2 \left(\epsilon_2 - D(\hat{p}(\mathbf{x}|y), p_\theta(\mathbf{x}|y))\right)$$

In order to derive Equation 4, we restrict $\hat{p}$ to the Dirac-delta distribution generated by querying the learned inverse map $f_\theta^{-1}$ at a specific value of $z$. Now note that the first term in the Lagrangian corresponds to maximizing the "reconstructed" $\hat{y}$ similarly to the first term in Equation 4. If $p_f$ is assumed to be a Gaussian random variable with a fixed variance, then $\log p_f(\hat{y}|\mathbf{x}) = -||\hat{y} - \mu(\mathbf{x})||_2^2$, where $\mu$ is the mean of the probabilistic forward map.

Finally, in order to obtain the $\log p_0(z)$ term, note that, $D(\hat{p}(\mathbf{x}|y), p_\theta(\mathbf{x}|y)) \leq D(\delta_z(\cdot), p_0(\cdot)) \leq -\log p_0(z) + C$ (the first part follows by the data processing inequality for KL-divergence), where $C$ is a constant. To note why the second part of the previous equality follows, observe that $C = 0$, and equality holds when $z$ is discrete. When $z$ is continuous, if we define the width of the Dirac-delta, $\delta_z(\cdot)$ as a constant, $\alpha$, i.e. if

$$\delta_z(z_0) = \frac{1}{2\alpha} \cdot \mathbf{1}_{|z-z_0|\leq\alpha},$$

then the inequality holds with $C = -\log(2\alpha)$, which is a constant. Hence, constraining $\log p_0(z)$ instead of the true divergence gives us a lower bound on $\mathcal{L}$. Maximizing this lower bound (which is the same as Equation 4) hence also maximizes the true Lagrangian $\mathcal{L}$.

**How do we perform the optimization in Equation 4 in practice?**   Since Equation 4 describes a constrained optimization problem, our approach towards solving this problem in practice is via dual gradient descent. Gradient descent is used to optimize the Lagrangian of Equation 4 (with the constraint $p(\mathbf{z}) \geq \epsilon_2$ modified to be $\log p(\mathbf{z}) \geq \epsilon_2$ as it is easy to use $\log p(\mathbf{z})$ numerically for stochastic optimization), shown in Equation 5.

$$\mathcal{L}(y, \mathbf{z}; \lambda_1, \lambda_2) = f_\theta(f_\theta^{-1}(\mathbf{z}, y)) - \lambda_1 \left(||y - f_\theta(f_\theta^{-1}(\mathbf{z}, y))||_2^2 - \epsilon_1\right) + \lambda_2 \left(\log p(\mathbf{z}) - \epsilon_2\right) \qquad (5)$$

We now maximize the Lagrangian shown above with respect to $y$ and $\mathbf{z}$ using gradient descent. Ideally, the dual variables, $\lambda_1$ and $\lambda_2$ are supposed to be optimized together with $y$ and $\mathbf{z}$, however, we find it convenient to choose them to be fixed as a constant throughout, at $\lambda_1 = 10.0$ and $\lambda_2 = 0.5$, during training, analogous to penalty methods in constrained optimization. The same values of $\lambda_1$ and $\lambda_2$ are used for all domains in our experiments.

## B   Bias-Variance Tradeoff in MIN training

In this section, we provide details on the bias-variance tradeoff that arises in MIN training. Our analysis is primarily based on analysing the bias and variance in the $\ell_2$ norm of the gradient with respect

to the scenario where we are provided access to infinite samples from the distribution over optimal $y$s, $p^*(y)$ (this is a Dirac-delta distribution when function $f(\mathbf{x})$ evaluations are deterministic, and a distribution with non-zero variance when the function evaluations are stochastic or are corrupted by noise). Let $\hat{\mathcal{L}}_p(\mathcal{D}) = \frac{1}{|\mathcal{Y}|} \sum_{y_j \sim p_\mathcal{D}(y)} \frac{p(y_j)}{p_\mathcal{D}(y_j)} \left( \frac{1}{|N_{y_j}|} \sum_{k=1}^{|N_{y_j}|} \hat{D}(\mathbf{x}_{j,k}, f^{-1}(y_j)) \right)$ denote the empirical objective that the inverse map is trained with. We first analyze the variance of the gradient estimator and then analyse its bias. In order to analyse the variance, we will need the expression for variance of the importance sampling estimator, which is captured in the following Lemma.

**Lemma B.1** (Variance of IS [21]). *Let $P$ and $Q$ be two probability measures on the space $(\mathcal{X}, \mathcal{F})$ such that $d_2(P||Q) < \infty$. Let $\mathbf{x}_1, \cdots, \mathbf{x}_N$ be $N$ randomly drawn samples from $Q$, and $f : \mathcal{X} \to \mathbb{R}$ is a uniformly-bounded function. Then for any $\delta \in (0, 1]$, with probability atleast $1 - \delta$,*

$$\mathbb{E}_{x \sim P}[f(x)] \in \left[ \frac{1}{N} \sum_{i=1}^{N} w_{P/Q}(x_i) f(x_i) \pm ||f||_\infty \sqrt{\frac{(1 - \delta) d_2(P||Q)}{\delta N}} \right] \tag{6}$$

Equipped with Lemma B.1, we are ready to derive an expression for the variance in the gradient due to reweighting to a distribution for which only a few datapoints are observed.

**Lemma B.2** (Gradient Variance Bound for MINs). *Let the inverse map be given by $f_\theta^{-1}$. Let $N_y$ denote the number of datapoints observed in $\mathcal{D}$ with score equal to $y$, and let $\hat{\mathcal{L}}_p(\mathcal{D})$ be as defined above. Let $\mathcal{L}_p(p_\mathcal{D}) = \mathbb{E}[\hat{\mathcal{L}}_p(\mathcal{D})]$, where the expectation is computed with respect to the dataset $\mathcal{D}$. Assume that $||\nabla_\theta \hat{D}(\mathbf{x}, f^{-1}(y))||_2 \leq L$ and $\mathrm{var}[\nabla_\theta \hat{D}(\mathbf{x}, f^{-1}(y))] \leq \sigma^2$. Then, there exist some constants $C_1, C_2$ such that with a confidence at least $1 - \delta$,*

$$\mathbb{E}\left[ ||\nabla_\theta \hat{\mathcal{L}}_p(\mathcal{D}) - \nabla_\theta \mathcal{L}_p(p_\mathcal{D})||_2^2 \right] \leq C_1 \mathbb{E}_{y \sim p(y)} \left[ \sigma^2 \frac{\log \frac{1}{\delta}}{N_y} \right] + C_2 L^2 \frac{(1 - \delta) d_2(p||p_\mathcal{D})}{\delta \sum_{y \in \mathcal{D}} N_y}$$

*Proof.* We first bound the range in which the random variable $\nabla_\theta \hat{\mathcal{L}}_p(\mathcal{D})$ takes values in as a function of number of samples observed for each $y$. All the steps follow with high probability, i.e. with probability greater than $1 - \delta$,

$$\nabla_\theta \hat{\mathcal{L}}_p(\mathcal{D}) = \nabla_\theta \frac{1}{|\mathcal{Y}_\mathcal{D}|} \sum_{y_j \sim p_\mathcal{D}(y)} \frac{p(y_j)}{p_\mathcal{D}(y_j)} \cdot \left( \frac{1}{|N_{y_j}|} \sum_{k=1}^{|N_{y_j}|} \hat{D}(\mathbf{x}_{j,k}, f^{-1}(y_j)) \right)$$

$$\in \frac{1}{|\mathcal{Y}_\mathcal{D}|} \sum_{y_j \sim p_\mathcal{D}(y)} \left[ \mathbb{E}_{\mathbf{x}_{ij} \sim p(\mathbf{x}|y_j)} \left[ \hat{D}(\mathbf{x}_{ij}, y_j) \right] \pm \sqrt{\frac{\mathrm{var}(\hat{D}(x, y)) \cdot (\log \frac{!}{\delta})}{\delta \cdot N_y}} \right]$$

$$\in \mathbb{E}_{y_j \sim p(y)} \left[ \mathbb{E}_{\mathbf{x}_{ij} \sim p(\mathbf{x}|y_j)} \left[ \hat{D}(\mathbf{x}_{ij}, y_j) \right] \pm \sqrt{\frac{\mathrm{var}(\hat{D}(x, y)) \cdot (\log \frac{!}{\delta})}{\delta \cdot N_y}} \right] \pm \sqrt{\frac{(1 - \delta) \cdot d_2(p(y)||p_\mathcal{D}(y))}{\delta \cdot \sum_{y_j \in \mathcal{D}} N_{y_j}}} \tag{7}$$

where $d_2(p||q)$ is the exponentiated Renyi-divergence between the two distributions $p$ and $q$, given by, $d_2(p(y)||q(y)) = \int_y q(y) \left( \frac{p(y)}{q(y)} \right)^2 dy$. The first step follows by applying Hoeffding's inequality on each inner term in the sum corresponding to $y_j$ and then bounding the variance due to importance sampling $y$s finally using concentration bounds on variance of importance sampling using Lemma B.1.

Thus, the gradient may lie in the entire range of values as defined above with high probability. Thus, with high probability, atleast $1 - \delta$,

$$\mathbb{E}\left[ ||\nabla_\theta \hat{\mathcal{L}}_p(\mathcal{D}) - \nabla_\theta \mathcal{L}_p(p_\mathcal{D})||_2^2 \right] \leq C_1 \mathbb{E}_{y \sim p(y)} \left[ \sigma^2 \frac{\log \frac{1}{\delta}}{N_y} \right] + C_2 L^2 \frac{(1 - \delta) d_2(p||p_\mathcal{D})}{\delta \sum_{\mathcal{Y}_\mathcal{D}} N_y} \tag{8}$$

and this concludes the variance result, and a proof of this Lemma. $\qquad \square$

**Bounding the bias.** The next step is to bound the bias in the gradient that arises due to training on a different distribution than the distribution of optimal $y$, $p^*(y)$. This can be written as follows:

$$||\mathbb{E}_{y \sim p^*(y)}[\mathbb{E}_{\mathbf{x} \sim p(\mathbf{x}|y)}[D(\mathbf{x}, y)]] - \mathbb{E}_{y \sim p(y)}[\mathbb{E}_{\mathbf{x} \sim p(\mathbf{x}|y)}[D(\mathbf{x}, y)]]||_2^2 \leq \mathrm{D}_{\mathrm{TV}}(p, p^*)^2 \cdot L. \tag{9}$$

where $D_{TV}$ is the total variation divergence between two distributions $p$ and $p^*$, and L is a constant that depends on the maximum magnitude of the divergence measure $D$. Combining Lemma B.2 and the above result, we prove Theorem 3.1.

# C  Active Data Collection via Randomized Labeling

In this section, we explain in more detail the randomized labeling algorithm described in Section 3.4. Before going onto the derivation, we provide a more detailed intuitive explanation of the method and how it relates to posteriors and Thompson sampling. After than we present a derivation that proves a regret bound.

## C.1  Detailed Intuitive Explanation of Randomized Labeling

While the passive setting requires care in finding the best value of $y$ for the inverse map, the active setting presents a different challenge: choosing a new query point $\mathbf{x}$ at each iteration to augment the dataset $\mathcal{D}$ and make it possible to find the best possible optimum. Prior work on bandits and Bayesian optimization often uses Thompson sampling (TS) [30, 31, 36] as the data-collection strategy. TS maintains a posterior distribution over functions $p(f_t|\mathcal{D}_{1:t})$. At each iteration, it samples a function from this distribution and queries the point $\mathbf{x}_t^\star$ that greedily minimizes this function. TS offers an appealing query mechanism, since it achieves sub-linear Bayesian regret (the expected cumulative difference between the value of the optimal input and the selected input), given by $\mathcal{O}(\sqrt{T})$, where $T$ is the number of queries.

Maintaining a posterior over high-dimensional parametric functions is generally intractable. However, we can approximate Thompson sampling with MINs. First, note that sampling $f_t$ from the posterior is equivalent to sampling $(\mathbf{x}, y)$ pairs consistent with $f_t$ – given sufficiently many $(\mathbf{x}, y)$ pairs, there is a unique smooth function $f_t$ that satisfies $y_i = f_t(\mathbf{x}_i)$. For example, we can infer a quadratic function exactly from three points. For a more formal description, we refer readers to the notion of eluder dimension [29]. Thus, instead of maintaining intractable beliefs over the function, we can identify a function by the samples it generates, and define a way to sample synthetic $(\mathbf{x}, y)$ points such that they implicitly define a unique function sample from the posterior.

To apply this idea to MINs, we train the inverse map $f_{\theta_t}^{-1}$ at each iteration $t$ with an *augmented* dataset $\mathcal{D}_t' = \mathcal{D}_t \cup \mathcal{S}_t$, where $\mathcal{S}_t = \{(\tilde{\mathbf{x}}_j, \tilde{y}_j)\}_{j=1}^K$ is a dataset of synthetically generated input-score pairs corresponding to unseen $y$ values in $\mathcal{D}_t$. While there are many ways to generate these points, we simply sample $\mathbf{x}$ values randomly from $\mathcal{D}_t$, and sample $y$ values from $p^*(y)$, with positive additive noise. Training $f_{\theta_t}^{-1}$ on $\mathcal{D}_t'$ corresponds to training $f_{\theta_t}^{-1}$ to be an approximate inverse map for a function $f_t$ sampled from $p(f_t|\mathcal{D}_{1:t})$, as the synthetically generated samples $\mathcal{S}_t$ implicitly induce a model of $f_t$. We can then approximate Thompson sampling by obtaining $\mathbf{x}_t^\star$ from $f_{\theta_t}^{-1}$, labeling it via the true function, and adding it to $\mathcal{D}_t$ to produce $\mathcal{D}_{t+1}$, as shown in Algorithm 2 in Section 3.4.

## C.2  Regret Bound Proof for Randomized Labeling

Now, we prove regret bounds for the randomized labeling scheme in Algorithm 2. We first revisit Thompson sampling, then provide arguments for how our randomized labeling algorithm relates to it, highlight the differences, and then prove a regret bound for this scheme under mild assumptions for this algorithm. Our proof follows commonly available proof strategies for Thompson sampling.

---
**Algorithm 3** Thompson Sampling (TS)

---
1: Initialize a policy $\pi_a : \mathcal{X} \to \mathbb{R}$, data so-far $\mathcal{D}_0 = \{\}$, a prior over $\theta$ in $f_\theta - P(\theta^*|\mathcal{D}_0)$
2: **for** step $t$ in $\{0, \ldots, \text{T-1}\}$ **do**
3:     $\theta_t \sim P(\theta^*|\mathcal{F}_t)$     (Sample $\theta_t$ from the posterior)
4:     Query $\mathbf{x}_t = \arg\max_{\mathbf{x}} \mathbb{E}[f_{\theta_t}(\mathbf{x}) \mid \theta^\star = \theta_t]$ (Query based on the posterior probability $\mathbf{x}_t$ is optimal)
5:     Observe outcome: $(\mathbf{x}_t, f(\mathbf{x}_t))$
6:     $\mathcal{D}_{t+1} = \mathcal{D}_t \cup (\mathbf{x}_t, f(\mathbf{x}_t))$
7: **end for**

---

**Notation**  The TS algorithm queries the true function $f$ at locations $(\mathbf{x}_t)_{t \in \mathbb{N}}$ and observes true function values at these points $f(\mathbf{x}_t)$. The true function $f(\mathbf{x})$ is one of many possible functions that can be defined over the space $\mathbb{R}^{|\mathcal{X}|}$. Instead of representing the true objective function as a point

object, it is common to represent a distribution $p^*$ over the true function $f$. This is justified because, often, multiple parameter assignments $\theta$, can give us the same overall function. We parameterize $f$ by a set of parameters $\theta^*$.

The $T$ period regret over queries $\mathbf{x}_1, \cdots, \mathbf{x}_T$ is given by the random variable

$$\text{Regret}(T) := \sum_{t=0}^{T-1} [f(\mathbf{x}^\star) - f(\mathbf{x}_t)]$$

Since selection of $\mathbf{x}_t$ can be a stochastic, we analyse **Bayes risk** [30, 31], which is defined as the expected regret over randomness in choosing $\mathbf{x}_t$, observing $f(\mathbf{x}_t)$, and over the prior distribution $P(\theta^*)$. This definition is consistent with Russo and Van Roy [30].

$$\mathbb{E}[\text{Regret}(T)] = \mathbb{E}\left[\sum_{t=0}^{T-1} [f(\mathbf{x}^\star) - f(\mathbf{x}_t)]\right]$$

Let $\pi^{\text{TS}}$ be the policy with which Thompson sampling queries new datapoints. We do not make any assumptions on the stochasticity of $\pi^{\text{TS}}$, therefore, it can be a stochastic policy in general. However, we make 2 assumptions (A1, A2) to simplify analysis as used in Russo and Van Roy [30].

**A1:** $\sup_{\mathbf{x}} f(\mathbf{x}) - \inf_{\mathbf{x}} f(\mathbf{x}) \leq 1$ (Difference between max and min scores is bounded by 1) – If this is not true, we can scale the function values so that this becomes true.

**A2:** Effective size of $\mathcal{X}$ is finite. [1]

TS (Alg 3) queries the function value at $\mathbf{x}$ based on the posterior probability that $\mathbf{x}$ is optimal. More formally, the distribution that TS queries $\mathbf{x}_t$ from can be written as: $\pi_t^{\text{TS}} = \mathbb{P}(\mathbf{x}^* = \cdot|\mathcal{D}_t)$. When we use parameters $\theta$ to represent the function parameter, and thus this reduces to sampling an input that is optimal with respect to the current posterior at each iteration: $\mathbf{x}_t \in \arg\max_{\mathbf{x} \in \mathcal{X}} \mathbb{E}[f_{\theta_t}(\mathbf{x})|\theta^* = \hat{\theta}_t]$.

MINs (Alg 2) train inverse maps $f_\theta^{-1}(\cdot)$, parameterized as $f_\theta^{-1}(z, y)$, where $y \in \mathbb{R}$. We call an inverse map *optimal* if it is uniformly optimal given $\theta_t$, i.e. $||f_{\theta_t}^{-1}(\max_{\mathbf{x}} f(\mathbf{x})|\theta_t) - \delta\{\arg\max_{\mathbf{x}} \mathbb{E}[f(\mathbf{x})|\theta_t]\}|| \leq \varepsilon_t$, where $\varepsilon_t$ is controllable (usually the case in supervised learning, errors can be controlled by cross-validation).

Now, we are ready to show that the regret incurred the randomized labelling active data collection scheme is bounded by $\mathcal{O}(\sqrt{T})$. Our proof follows the analysis of Thompson sampling presented in Russo and Van Roy [30]. We first define *information ratio* and then use it to prove the regret bound.

**Information Ratio** Russo and Van Roy [30] related the expected regret of TS to its expected information gain i.e. the expected reduction in the entropy of the posterior distribution of $\mathcal{X}^*$. Information ratio captures this quantity, and is defined as:

$$\Gamma_t := \frac{\mathbb{E}_t \left[f(\mathbf{x}_t) - f(\mathbf{x}^\star)\right]^2}{I_t \left(\mathbf{x}^*; (\mathbf{x}_t, f(\mathbf{x}_t))\right)}$$

where $I(\cdot, \cdot)$ is the mutual information between two random variables and all expectations $\mathbb{E}_t$ are defined to be conditioned on $\mathcal{D}_t$. If the information ratio is small, Thompson sampling can only incur large regret when it is expected to gain a lot of information about which $\mathbf{x}$ is optimal. Russo and Van Roy [30] then bounded the expected regret in terms of the maximum amount of information any algorithm could expect to acquire, which they observed is at most the entropy of the prior distribution of the optimal $\mathbf{x}$.

**Lemma C.1 (Bayes-regret of vanilla TS)[30]).** *For any $T \in \mathbb{N}$, if $\Gamma_t \leq \overline{\Gamma}$ (i.e. information ratio is bounded above) a.s. for each $t \in \{1, \ldots, T\}$,*

$$\mathbb{E}[Regret(T, \pi^{\text{TS}})] \leq \sqrt{\overline{\Gamma} H\left(\mathcal{X}^*\right) T}$$

We refer the readers to the proof of Proposition 1 in Russo and Van Roy [30]. The proof presented in Russo and Van Roy [30] does not rely specifically on the property that the query made by the Thompson sampling algorithm at each iteration $\mathbf{x}_t$ is posterior optimal, but rather it suffices to have a bound on the maximum value of the information ratio $\Gamma_t$ at each iteration $t$. Thus, if an algorithm chooses to query the true function at a datapoint $\mathbf{x}_t$ such that these queries always contribute in learning more about the optimal function, i.e. $I(\cdot, \cdot)$ appearing in the denominator of $\Gamma$ is always more than a threshold, then information ratio is lower bounded, and that active data collection algorithm will have a sublinear asymptotic regret. We are interested in the case when the active data collection algorithm queries a datapoint $\mathbf{x}_t$ at iteration $t$, such that $\mathbf{x}_t$ is the optimum for a function $\hat{f}_{\hat{\theta}_t}$, where $\hat{\theta}_t$ is a sample from the posterior distribution over $\theta_t$, i.e. $\hat{\theta}_t$ lies in the high confidence region of the posterior distribution over $\theta_t$ given the data $\mathcal{D}_t$ seen so far. In this case, the mutual information between the optimal datapoint $\mathbf{x}^\star$ and the observed $(\mathbf{x}_t, f(\mathbf{x}_t))$ input-score pair is likely to be greater than 0. More formally,

$$I_t(\mathbf{x}^\star, (\mathbf{x}_t, f(\mathbf{x}_t))) \geq 0 \quad \forall \; \mathbf{x}_t = \arg\max_{\mathbf{x}} f_{\hat{\theta}_t}(\mathbf{x}) \; \text{ where } \; P(\hat{\theta}_t | \mathcal{D}_t) \geq \epsilon_{\text{threshold}} \qquad (10)$$

The randomized labeling scheme for active data collection in MINs performs this step. The algorithm samples a bunch of $(\mathbf{x}, y)$ datapoints, sythetically generated, – for example, in our experiments, we add noise to the values of $\mathbf{x}$, and randomly pair them with unobserved or rarely observed values of $y$. If the underlying true function $f$ is smooth, then there exist a finite number of points that are sufficient to uniquely describe this function $f$. One measure to formally characterize this finite number of points that are needed to uniquely identify all functions in a function class is given by *Eluder dimension* [29].

By augmenting synthetic datapoints and training the inverse map on this data, the MIN algorithm ensures that the inverse map is implicitly trained to be an accurate inverse for the unique function $f_{\hat{\theta}_t}$ that is consistent with the set of points in the dataset $\mathcal{D}_t$ and the augmented set $\mathcal{S}_t$. Which sets of functions can this scheme represent? The functions should be consistent with the data seen so far $\mathcal{D}_t$, and can take randomly distributed values outside of the seen datapoints. This can roughly argued to be a sample from the posterior over functions, which Thompson sampling would have maintained given identical history $\mathcal{D}_t$.

**Lemma C.2 (Bounded-error training of the posterior-optimal $\mathbf{x}_t$ preserves asymptotic Bayes-regret).** *$\forall t \in \mathbb{N}$, let $\hat{\mathbf{x}}_t$ be any input such that $f(\hat{\mathbf{x}}_t) \geq \max_{\mathbf{x}} \mathbb{E}[f(\mathbf{x})|\mathcal{D}_t] - \varepsilon_t$. If MIN chooses to query the true function at $\hat{\mathbf{x}}_t$ and if the sequence $(\varepsilon_t)_{t \in \mathbb{N}}$ satisfies $\sum_{t=0}^{T} \varepsilon_t = \mathcal{O}(\sqrt{T})$, then, the regret from querying this $\varepsilon_t$-optimal $\hat{\mathbf{x}}_t$ which is denoted in general as the policy $\hat{\pi}^{\text{TS}}$ is given by $\mathbb{E}[Regret(T, \hat{\pi}^{\text{TS}})] = \mathcal{O}(\sqrt{T})$.*

*Proof.* This lemma intuitively shows that if posterior-optimal inputs $\mathbf{x}_t$ can be "approximately" queried at each iteration, we can still maintain sublinear regret. To see this, note:

$$f(\mathbf{x}^\star)) - f(\hat{\mathbf{x}}_t) = f(\mathbf{x}^\star) - f(\mathbf{x}_t) + f(\mathbf{x}_t) - f(\hat{\mathbf{x}}_t).$$

and hence, we can simplify:

$$\mathbb{E}[\text{Regret}(T, \hat{\pi}^{\text{TS}})] = \mathbb{E}[\text{Regret}(T, \pi^{\text{TS}})] + \mathbb{E}[\sum_{t=1}^{T}(f(\mathbf{x}_t) - f(\hat{\mathbf{x}}_t))]$$

The second term can be bounded by the absolute value in the worst case, which amounts $\sum_{t=0}^{T} \varepsilon_t$ extra Bayesian regret. As Bayesian regret of TS is $\mathcal{O}(\sqrt{T})$ and $\sum_{t=0}^{T} \varepsilon_t = \mathcal{O}(\sqrt{T})$, the new overall regret is also $\mathcal{O}(\sqrt{T})$. $\qquad\square$

**Theorem C.3 (Bayesian Regret of randomized labeling active data collection scheme proposed in Section 3.4 is $\mathcal{O}(\sqrt{T})$).** *Regret incurred by the MIN algorithm with randomized labeling is of the order $\mathcal{O}(\sqrt{(\bar{\Gamma} H(\mathcal{X}^*) + C)T})$.*

*Proof.* Simply put, we will combine the insight about the mutual information $I(\mathbf{x}^\star, (\mathbf{x}_t, f(\mathbf{x}_t))) > 0$ and C.2 in this proof. Non-zero mutual information indicates that we can achieve a $\mathcal{O}(\sqrt{T})$ regret if we query $\mathbf{x}_t$s which are optimal corresponding to some implicitly defined forward function lying in

Figure 4: Optimized images produced by a cGAN model for the youngest face optimization task on the IMDB-faces dataset. We observe that the cGAN model ignores the score value and produces similar images to an unconditional model, without any noticeable correlation with the score value. The samples produced mostly correspond to the most frequently occurring images in the dataset. Compare to the images shown in Figure 3, and note the clear difference in the apparent ages.

the high confidence set of the true posterior given the observed datapoints $\mathcal{D}_t$. Lemma C.2 says that if bounded errors are made in fitting the inverse map, the overall regret remains $\mathcal{O}(\sqrt{T})$.

More formally, if the following is true: $||f_{\theta_t}^{-1}(\max_{\mathbf{x}} f(\mathbf{x})|\theta_t) - \delta\{\arg\max_{\mathbf{x}} \mathbb{E}[f(\mathbf{x})|\theta_t]\}|| \leq \delta_t$, then,

$$||\mathbb{E}_{\mathbf{x}_t \sim f_{\theta_t}^{-1}}[f(\mathbf{x}_t)] - \mathbb{E}_{\mathbf{x}'_t \sim \pi_t^{\mathrm{TS}}}[f(\mathbf{x}'_t)]|| \leq ||f(\cdot)||_\infty \cdot ||f_{\theta_t}^{-1} - \pi_t^{\mathrm{TS}}|| \leq \delta_t R_{\max} \leq \varepsilon_t \quad (11)$$

and now application of Lemma C.2 gives us the extra regret incurred. (Note that this also provides us a way to choose the number of training steps for the inverse map)

Further, note if we sample $\mathbf{x}_t$ at iteration $t$ from a distribution that shares support with the true posterior over optimal $\mathbf{x}_t$ (which is used by TS), we still incur sublinear, bounded $\mathcal{O}(\sqrt{\bar{\Gamma}H(A^*)T})$ regret.

In the worst case, the overall bias caused due to the approximations will lead to an additive cumulative increase in the Bayesian regret, and hence, there is a constant $\exists\ C \geq 0$, such that $\mathbb{E}[\mathrm{Regret}(T, f^{-1})] = \mathcal{O}(\sqrt{(\bar{\Gamma}H(\mathcal{X}^*) + C)T})$. $\qquad\square$

# D  Additional Experiments and Experimental Details

In this appendix, we present some additional experiments, particularly on image inputs, and discuss the details of our experimental setup.

## D.1  Comparison to a standard cGAN baseline in Semantic Optimization over Face Images.

We compare our MIN model to a cGAN baseline on the IMDB-Wiki faces dataset for the semantic age optimization task shown in Table 2. In general, we found that the cGAN model learned to ignore the score value passed as input even when trained on the entire dataset (without excluding the youngest faces), and behaved similarly to a regular unconditional GAN model when queried to produce images $\mathbf{x}$ corresponding to the lowest age. We suspect that this could possibly be due to the fact that age of a person doesn't have enough direct signal to guide the model to utilize it effectively without re-weighting. We present the optimized images in Figure 4, which may be compared with images produced by MINs in Figure 3 in the main text.

## D.2  Experimental Details and Setup

In this section, we explain the experimental details and the setup of our model.

**Choice of base models/architectures.**    For our experiments involving MNIST image optimization and global optimization of benchmark functions task, we used the same architecture as a fully

connected GAN, where the generator and discriminator are both fully connected networks. We based our code for this part on an open-source implementation [19]. For the forward model experiments in these settings, we used a 3-layer feedforward ReLU network with hidden units of size 256 each in this setting. For all experiments on CelebA and IMDB-Wiki faces, we used the VGAN [24] model and the associated codebase as our starting setup. For experiments on batch contextual bandits, we used a fully connected discriminator and generator for MNIST, and a convolutional generator and ResNet18 discriminator for CIFAR-10. The prediction in this setting is categorical – 1 of 10 labels need to be predicted, so instead of using REINFORCE or derivative-free optimization to train the inverse map, we used the Gumbel-softmax [14] trick with a temperature $\tau = 0.75$, so as to use stochastic gradient descent to train the model. For the protein flourescence maximization experiment, we used a 2-layer, 256-unit feed-forward gumbel-softmax inverse map and a 2-layer feed-forward discriminator.

We trained models present in open-source implementations of BanditNet [32], but were unable to reproduce results as reported by Joachims et al. [15]. Thus we reported the paper reported numbers from the BanditNet paper in the main text as well as the performance we obtained.

**Hyperparameters for reweighting.**

- **Temperature hyperparameter** $\tau$, which is used to compute the reweighting distribution, according to the relationship: $p(b_i) \propto \frac{N_{b_i}}{N_{b_i}+\lambda} \exp\left(\frac{-|b_i-y^*|}{\tau}\right)$, is adaptively chosen based on the 90$^{\text{th}}$ percentile score in the dataset. For example, if the difference between $y_{max}$ and $y_{90^{\text{th}}-\text{percentile}}$ is given by $\alpha$, we choose $\tau = \alpha$. This scheme can adaptively change temperatures in the active setting.

- **Constant** $\lambda$ which decides whether the bin corresponding to a particular value of $y$ is small or not, we first convert the expression $\frac{N_y}{N_y+\lambda}$ to use densities rather than absolute counts, that is, $\frac{\hat{p}_{\mathcal{D}}(y)}{\hat{p}_{\mathcal{D}}(y)+\lambda}$, where $\hat{p}_{\mathcal{D}}(y)$ is the empirical density of observing $y$ in $\mathcal{D}$, and now we use the same constant $\lambda = 0.01$ for all the domains. We did not observe a lot of sensitivity to $\lambda$ values in the range $[0.003, 0.01]$, all of which performed reasonably similar.

- **Number of bins** was fixed to 20 for the purposed of reweighting, however we reiterate that the inverse map still needs to be trained on continuous $y$ values, which helps it extrapolate.

**Active data collection.** In practice, we train two copies of $f_\theta^{-1}$. The first, $f_{\text{expl}}^{-1}$, is trained with data augmented via synthetically generated samples (i.e., $\mathcal{D}'_t$). The other copy, $f_{\text{exploit}}^{-1}$, is trained on only real samples (i.e., $\mathcal{D}_t$). This improves stability during training – giving rise to more stable GAN optimization, which typically tends to be notoriously hard [20]. After training, we infer best possible $\mathbf{x}^\star$ from the trained model using the inference procedure described in Section 3.2.

### D.3 Visualization of Randomized Labeling Scheme on a didactic 1D-example

We also provide the visualization of the randomized labeling scheme on a 1D-function. The goal is to find the optimum of a function that takes a one dimensional input and produces a one dimensional output.

Visualization of the randomized labeling scheme is provided (as gifs) at this anonymous URL: `https://ibb.co/album/Jv4hxT` The groundtruth function is marked as a dashed line, and the queried points are marked in blue. Additionally the output of the MIN (with and without the Approx-Infer inference schemes) is shown as orange and green points respectively. Note that after more iterations of training, the output of the MIN converges to the maximum of the function.

We provide visualizations from three different initialization points, i.e. the function evaluations provided to the MIN learning algorithm initially. We also include the gifs in the supplementary material, but the link might be easier to navigate to.

## Footnotes

[1] By effective size we refer to the intrinsic dimensionality of $\mathcal{X}$. This doesn't necessarily imply that $\mathcal{X}$ should be discrete. For example, under linear approximation to the score function $f_\theta(\mathbf{x})$, i.e., if $f_\theta(\mathbf{x}) = \theta^T \mathbf{x}$, this defines a polyhedron but just analyzing a finite set of just extremal points of the polyhedron works out, thus making $|\mathcal{X}|$ effectively finite.