[Reviews · NeurIPS 2020]

Review 1

Summary and Contributions: The paper presents a method for model-based optimization. Instead of learning to predict the objective value y as a function of the input x (and using this to infer x*), the paper proposes to model x as a function of y. The mapping y -> x is one-to-many, so the "model inversion network" models a distribution over possible x given y. This is done by modeling xhat = NN( y, z ) where z is a random variable drawn from some predefined prior. A divergence between the model distribution (xhat given y) and the data distribution (x given y) is minimized; the experiments in the paper use the JS divergence (resulting in a model y -> xhat which looks like a conditional GAN). The paper introduces extra machinery to help make use of MINs: a method to use a model predicting y from x to decide when to trust the MIN, a reweighting method to train the MIN to be accurate in the high-objective-value areas where it will be used to search for x*, a Thompson sampling / randomized labeling method to actively collect data points. The experiments show strong performance on classification with bandit feedback, optimizing images, model-based optimization of controllers, and a protein design task.

Strengths: I like the main idea of the paper (which to my knowledge is novel). Thank you to the authors for writing and submitting this. The problem of "stepping off the manifold of realistic designs" is ubiquitous in model-based optimization. Learning a generative mapping from scores to designs is an elegant idea, and intuitively should avoid unrealistic designs. The paper is clearly written and easy to read. Experimental validation of the method is provided on a variety of challenging tasks, including optimization of images, model-based optimization of Gym controllers, and protein design. There is convincing experimental evidence that MINs can often produce good results for MBO.

Weaknesses: My main concern about the paper is that the proposed method has many moving parts. The method requires the following components: (i) Training a GAN (which is difficult and sensitive to hyperparameters). How many different random seed runs did you try for each expt: what was the variance? What was the sensitivity to hyperparameters? Did you observe issues with mode collapse (or is this not an issue here, because even if we don't generate all possible x, we might still hope to generate some *good* x?) (ii) Approx-Infer: training a model to predict yhat from x, and optimizing (y, z) to maximize the predicted score yhat(xhat(y, z)) while minimizing the disagreement between yhat and y. This involves (i) training another model and (ii) selecting three hyperparameters for minimizing a Lagrangian (eps2, lam1, lam2 in Appendix A). How do you choose these parameters in practice and how sensitive is the Lagrangian minimization to these parameters? Does it break if we choose eps too large, for example, and get overly conservative if eps is too small? What does it mean to do MINs without approx infer? - one still needs to pick y - it would seem maximizing y "without approx infer" would involve taking the limit of xhat as y -> infty, but this seems computationally impossible. (iii) Training distribution reweighting. What is K on line 187? I can't find it mentioned in the text anywhere else (and it seems like it will be crucial to shaping the reweighting distribution). I'm guessing it gets folded into lambda and tau in the later "practical implementation" (iv) Active data collection Do you re-initialize the nets and retrain after each round of active data collection? I think each of these components are well-motivated. However, I think the sheer amount of moving parts here may prohibit use of this method by the community in its current form (although a good library, or an extension which simplifies things while retaining the desirable properties, could help here). I also think some of the baselines could be stronger. In Table 2, the citation is to "Practical Bayesian optimization of machine learning algorithms", which uses GPs: did you use a GP or a NN for the baseline here? The text says the baseline was a "forward model of the form f_theta (x)" which has the implication of using an NN. If a GP was used, this seems inappropriate for these high-dimensional tasks. If a NN was used, the paper should compare to a method which uses uncertainty or otherwise tries to stay on the manifold of realistic designs. (Components (ii),(iii),(iv) of the submission are somewhat orthogonal to the MIN and so it seems unfair to compare to completely vanilla methods). Good options to look at here are "Model-Ensemble Trust-Region Policy Optimization", "Deep Bayesian Bandits Showdown", and their forward/backward cites. It would also be nice to see the reweighting and active data collection techniques combined with the baselines, as these seem entirely orthogonal to the MIN. I'm not familiar with the baselines in Table 1 / Table 3: it would be nice to have a description of these models in the text of the paper, so people can get an idea without having to follow the cite. In summary, the methods are appealing and well motivated, but there are a *lot* of moving parts, it's unclear exactly which components are responsible for which gains, and some of the baselines are a bit unclear / possibly could be stronger. Also the paper makes the claim that MIN's main purpose is to constrain MBO to realistic designs: it would be really nice to get more intuition into this, via theory and via (potentially toy) experiments / visualizations. I am slightly in doubt as to how much of "constraining to realistic designs" is being done by MIN and how much by Approx-Infer. The ablation suggests that Approx-Infer is critical to this, in which case a comparison to model-ensemble or uncertainty-based methods is necessary, as these methods (which are somewhat similar to Approx-Infer, in terms of constraining optimization to a trust region) also help constrain to realistic designs.

Correctness: Modulo comments above, the empirical methodology seems correct. I have not checked the proof of Thm 3.1 but it seems likely to be correct.

Clarity: The paper is well written and clear (modulo comments above). The appendix seems to have sufficient detail for implementing the method and has guidance for hyperparameters.

Relation to Prior Work: To my knowledge the proposed method is novel, and the related work section does a solid job discussing other contributions to MBO.

Reproducibility: Yes

Additional Feedback: --- Update after rebuttal: Thanks authors for the clarifications in the rebuttal and pointers to the ablation. Thanks also for running experiments vs a forward model ensemble (with and without variance penalty). I think this is useful for making your case. (I think in the camera ready you should include these comparisons for all of the experiments, with full experimental details of the forward model baseline, although I fully understand this is not reasonable to expect for the author response with both time and space constraints). I maintain my score: I think this is a good paper and should be accepted. --- Thanks for writing this paper. I think this is a neat and well executed idea, although I do wish it had fewer moving parts. There is some discussion of how to set hyperparameters in the appendix: I think this should be moved to or referenced from the main paper. The paper would be stronger with more done to understand the decoupled effects of (i) MINs vs predicting y from x, (ii) trust regions (approx-infer, model ensembles, uncertainty etc) vs no, (iii) reweighting the training distribution vs no, (iv) active data collection, as these modules all seem somewhat orthogonal (in their purpose, if not in specific implementation). Assigning a score for this paper, I am torn between liking the proposed method and the evidence that the method can produce some good results, and the lack of evidence that MINs produce realistic designs without approx-infer, the lack of evidence for the role of each component, and the lack of comparison to baselines with similar components. Therefore although I think I understood the paper, I have a somewhat diffuse posterior over what score I think the paper should get.


Review 2

Summary and Contributions: Thanks for addressing my questions and concerns in the author response. ---- This paper proposes "model inversion networks" (MINs), a method of maximizing a black-box function f(x), with a specific focus on learning from logged data (x,f(x)), although active learning is also considered. In addition, they focus on the setting of a high-dimensional input space for which "valid inputs" are only a small subset of this space. One of the main ideas of this work is to construct a generative model for inputs x that achieve a given score y. At the same time, a "forward model" is built to approximate f(x) directly. The idea is that by restricting our search to x's proposed by the inverse generative model (for large values of y), we're more likely to stay in the distribution of valid inputs, and that if there is a large gap between the predicted f(x) and the value of y we started with (that we used to seed the inverse generative model), then we should avoid that x because something is off with our models in that part of the input space. The actual implementation of these models differs somewhat from this high-level description, and I'll revisit with some specific questions below. The paper finds an advantage from reweighting the training data by importance sampling to a particular distribution that they derive from a bound in a theorem. They also describe a method to supplement the logged training data by active queries.

Strengths: Tackled a challenging problem that should be of interrest to the NeurIPS community. Main ideas and motivations are explained well. Experimented with several different problems.

Weaknesses: The problem statement definition is rather vague, perhaps intentionally so: find the x that attains the highest possible value of f(x). Perhaps it's too advanced at this point, but the way performance is measured should effect how we choose x's. For Theorem 3.1, I think some more justification should be given for the relevance of the MSE between the gradient of the objective function on the data and on the distribution centered at the optimal y. Specifically, why the gradient as opposed to the objective itself, and is MSE strong enough for how you want to use it? I guess it's nice that you can derive a distribution p that balances the bias/variance tradeoff for the bound... but without some more explanation, it feels like a stretch for this to be the right probability to use in practice. Very possible that I'm missing some intuition here and it should be more clear to me. The objective function for prediction given between lines 162 and 163 seems reasonable enough. But something seems missing on the objective function for learning the various models. We're looking for Disc(x|y), which is looking for an x that is both a valid input AND y is the score for x. However, in the objective function, it seems that the only negative examples that Disc is trained on are those for which x is generated. It's not obvious that Disc will have any reason to look at y at all -- on the training data, it should be sufficient to figure out whether or not x is generated. Would it make sense to have a term in which true x's are given with incorrect y's? The having a 'forward model' that can correctly predict the y seems important to the objective function between liunes 162 and 163.

Correctness: I don't see anything wrong, per se, just some clarity issues as described below.

Clarity: - I think some additional explanation could be provided for the basic experiment in Batch contextual bandits subsection. The model setup is clear, but does "testing" amount to seeding a 1 for the label y, and seeing if a randomly drawn label x is correct? - For character stroke width optimization, the definition of this problem is not clear to me. Are you measuring "recognizability" by some fixed model trained on MNIST? And is recognizable defined as whether or not the class with the highest probability is the correct one? How is stroke width measured?

Relation to Prior Work: Seems reasonable.

Reproducibility: Yes

Additional Feedback: Interesting work on attacking a very tough problem.


Review 3

Summary and Contributions: This paper introduces model inversion networks as a solution to practical data-driven optimization problems. The core idea of the proposed solution is that of a stochastic inverse mapping from rewards to data points. To ensure the generation of in-distribution data points the authors propose a clever coupling of their inverse mapping with an independently-trained forward model. The authors demonstrate that their method goes beyond the generation of data points to active data collection. The claims of the paper are also validated experimentally.

Strengths: The proposed framework presents a very practical and intuitive take on model-based optimization. The approach that the authors put forward (built around the concept of stochastic inverse maps) is elegant and the authors address a big challenge in using inverse maps, that of obtaining in-distribution samples while extrapolating, that via simple, yet insightful "trick": couple the inverse map with a forward network. Admittedly, I am not an expert in this area, but I find this trick very interesting. Given the simplicity of this trick, I would suspect that similar approaches have been considered in different contexts. Yet, I find its use in model-based optimization is insightful. The additional feature of reweighing and the use of a GAN to constrain the possible inputs are clean ideas. While the contribution is comprised by basic (admittedly not technically novel components) I find the approach elegant and for this reason I believe it has merit.

Weaknesses: My main criticism is around the experimental evaluation of this work. In several parts, the authors omit details on the experimental setup and leave it to the reader to investigate cited works. For instance, in Section 4.1: "Following Joachims et al. [15], we evaluate learned policies trained on a static dataset for simulated classification tasks on MNIST and CIFAR." This description is not self-contained and thus hard for the reader to fully appreciate all the details of this experiment. Similarly for the competing methods: the authors only namedrop them and do not provide any intuition as to what is the main difference between them and MINs. In general, the experimental evaluation section is weak. It lacks details of the experimental setup, the description of competing methods is subpar, and in some cases the results can be interpreted as subjective (e.g., in the case of semantic image optimization).

Correctness: The technical claims of this work are solid. The experimental evaluation of this work needs improvement as many details on the setting and the competing methods are omitted.

Clarity: The paper (apart from the experimental section) is very well written and easy to follow.

Relation to Prior Work: The papers presents a detailed comparison to prior works with respect to the functionalities the proposed approach. I would like to see a detailed comparison with respect to technical contributions. As mentioned in my comments above, the proposed work seems to build upon existing ideas (yet put together to form a novel framework).

Reproducibility: Yes

Additional Feedback: After author response: The authors have addressed several of my concerns. I am in favor of acceptance.


Review 4

Summary and Contributions: This paper targets to solve optimization problem that select input x to maximize unknow reward function. The authors propose model inversion networks, where it learned stochastic mapping from score/label y along with low-dimension latent representation to input x. The practical implementation is similar to a conditional GAN. Given the learned inverse maps, the inference (finding the optimal x) with constraints are done by solving a Lagrangian. Also, the authors reweighting and discretizing a sampled data point such that the inverse map tends to produce high value of y. The paper performs empirical evaluations on a lot of tasks, to show MINs can learn inverse map well, and different sub-components (e.g. exploration strategy, inference strategy) work well.

Strengths: According to section 4.1, the proposed approach (MINs) shows clear benefits over previously proposed model (e.g. BandiNet) in the data-driven contextual setting, and can outperform Proxy forward baseline in non-contextual setting on high-dimensional data task. According to section 4.2, MINs also shows better/comparable performance in activate data collecting setting comparing to previous methods. The paper also shows a bound of objective after reweighting the training distribution, which is the tradeoff between reducing variance by covering data distribution, the choice of p(y) and the actual optimal y.

Weaknesses: In real scenario, a data x can be associated with many characteristics. It seems that, current algorithm and evaluation mostly focusing on (x,y) where y is a scalar or single property. Is it possible to address the scenario such that score/label y is also multidimensional?

Correctness: Yes

Clarity: The paper is quite dense. It describes quite a lot of technical details, and similarly to the experimental section.

Relation to Prior Work: The related work looks concise and self-contained. The authors motivated their work very well, and clearly summarizes the contribution again in the discussion section.

Reproducibility: Yes

Additional Feedback: After reading authors' feedback and other reviewers' comment, I incline to keep my score the same as my initial review.

[Author Response · NeurIPS 2020]

We thank the reviewers for detailed comments and a positive assessment of our work. We will release a modularized
implementation of MINs alongside all the tasks to make it easy to use and reproduce the results **(R1)**. We will
incorporate experimental details into the main text and substantially revise the experimental section to clearly define
the baseline methods and the task setup **(R2, R3)**. We thank **R4** for proposing the interesting idea to extend MINs to
optimize multiple objectives, and we will discuss this as potential future work.

**R2: It's not obvious that Disc will have any reason to look at y at all:** The design of the conditional discriminator
is based directly on conditional GANs (cGANs) (Mirza et al. 2014), and should work for the same reason that cGANs
work. We will clarify this in the paper. Intuitively, the learning dynamics of a MIN (or a cGAN) would force the
generator to first produce valid samples, since the discriminator can otherwise easily distinguish fake and real samples
simply based on validity. Once the generator produces valid samples, the discriminator can only win in the GAN game
by exploiting the correlation between the conditional value $y$ and the output $x$, thus forcing the generator to accurately
model the conditional distribution $p(x|y)$. We experimented with the stategy R2 mentioned, that pairs real $x$ with
incorrect $y$s as negative examples, similar to Reed et al. 2016 (AC-GAN), but did not observe a benefit, and so we
omitted it in the interest of simplicity.

**R2, R3: Experimental evaluation details:** We will clearly define the task specifications and prior methods in the final
version of the paper in an extra page devoted to specifically this, and ensure that all experimental details are provided.
We already provide the hyperparameter settings and setup details in App. D.2, but will bring them to the main text and
add more details. To clarify some specific details: **(i)** the contextual bandit task requires predicting the right class label
$(x)$ of an image (which is the context) given access to $(c, x, y)$ tuples where $y = \mathbb{I}(\text{x is the correct label for image c})$.
The evaluation metric is simply the standard test classification accuracy for unseen test images. **(ii)** In stroke width
optimization, we define stroke width as the number of pixels with intensity more than a threshold (= 0.2) in the image.
An image is invalid if the number of pixels with intensity more than 0.2 is more than 90%, indicating that this image is
unlike MNIST digits. **(iii)** We provide **3 tasks** (controller, protein, and bandit) with quantitative and objective evaluation
scores. Two of these (protein, bandit) are taken directly from and compare to baselines from prior works.

**R2: Intuition for re-weighting distribution:** We already provide some intuition in 188-189, and we will elaborate
on this further here and in the paper. Our choice of $p$ weighs the higher $y$ values higher (thus reducing "bias" as a
result of training on suboptimal $y$ values although the inverse map is queried at only large $y$ values) while also ensuring
that the number of points for the chosen large $y$ values is large enough (thus reducing "variance"). The expression
for $p$ captures this tradeoff: $\exp(y - y^*)$ captures the bias, and $\frac{N_y}{N_y + K}$ (which becomes 0 for small $N_y$ and tends to 1
otherwise) captures the variance effect, and our choice of $p$ multiplies the two expressions.

**R1: Table 2 and new experimental comparisons:** We used a NN baseline here, and not a GP. We have the edited
the citation to include several other papers that utilize a forward function for optimization. We have also added a
comparison to **(1)** forward model ensemble + optimizing mean over the ensemble and, **(2)** forward model ensemble +
variance penalty (that captures uncertainty), and on the controller task, we find that over 4 seeds each, **(1)** attains a mean
return of **325.3** and **(2)** attains **165.3**, and MINs outperform both methods attaining **1960.1** mean return (see Table 2,
Hopper). We will add a comparison of these forward model variants to all the tasks in the final.

**R1: Details about the method.** We have included these details in the paper now. We respond here to some questions:
- **Decoupled effects of components / Ablations:** Lines 340-356 in Section 4.1 present ablation studies, where we study
the effect of decoupling **(i)** training the inverse map **(ii)** re-weighting and **(iii)** approx-infer. We have now presented
these results as a table more effectively in the paper. There is a small drop in performance w/o Approx-Infer (Table 1,
Fig 2, lines 340-353). If the reviewer suggests some new ablation studies, we are happy to add those in the final version.
- **Training a GAN:** We trained 4 seeds for the controller, protein, MNIST and bandit tasks and were able to run only
1 seed for face optimization, since it requires about a week to train. We directly used implementations of GANs
(details in App. D.2, lines 766-778) that have been previously employed on these domains *without* changing GAN
hyperparameters at all. We did observe some mode collapse, but found that the models were still able to produce good
solutions – this seems logical, since we are only interested in one mode in optimization (the one near the best solution).
- **Approx-Infer.** As discussed in lines 582-585 (App. A), we found it empirically convenient to use a fixed "penalty"
version, with $\lambda_1 = 10$ and $\lambda_2 = 0.5$. We fixed these values across all tasks and added a discussion of this in Sec. 3.2.
- **Practical $\lambda$ and $\tau$.** We used empirical densities and replace this choice by $\lambda$ and $\tau$ (discussed in App. D.2 (lines
783-792)), i.e., Thm 3.1 provides a functional form of the re-weighting distribution.
- **Exploration.** We did *not* re-initialize the nets after a round as this was computationally and performance-wise better.

**R2: Theorem 3.1 and MSE:** We have added a discussion of this in the paper now. We chose the gradient $\nabla_\theta \mathcal{L}(\mathcal{D})$,
since this gradient is used to update the parameters $\theta$ of the inverse map, and we are interested in choosing $p$ such that
the expected gradient under $p$ aligns close to the expected gradient on the optimal datapoint(s). The choice of MSE
between gradients was convenient and gave us a way to select the re-weighting, but in principle, we could use other
metrics, such as, finding the optimal $p$ by minimizing the cosine similarity between these gradients.

[Meta-Review · NeurIPS 2020]

All reviewers agree this is an interesting paper that should be accepted at NeurIPS. The paper tackles a challenging problem proposing a novel, elegant and insightful approach. The authors have committed to address the main concerns raised by the reviewers by including new baseline experiments, giving more details for the experimental evaluation and providing more details and clearer explanations of the model.